# A Hybrid Deep Learning Approach for ECG-Based Arrhythmia Classification

**DOI:** 10.3390/bioengineering9040152

**Published:** 2022-04-02

**Authors:** Parul Madan, Vijay Singh, Devesh Pratap Singh, Manoj Diwakar, Bhaskar Pant, Avadh Kishor

**Affiliations:** 1Department of Computer Science and Engineering, Graphic Era Deemed to Be University, Dehradun 248002, India; vijaysingh.cse@geu.ac.in (V.S.); devesh.geu@gmail.com (D.P.S.); manoj.diwakar@gmail.com (M.D.); bhasker.pant@geu.ac.in (B.P.); 2Department of Computer Science and Engineering, Thapar Institute of Engineering and Technology, Patiala 147004, India; avadh.kishor@thapar.edu

**Keywords:** arrhythmia, deep learning, ECG, classification, convolutional neural network (CNN), long short-term memory (LSTM)

## Abstract

Arrhythmias are defined as irregularities in the heartbeat rhythm, which may infrequently occur in a human’s life. These arrhythmias may cause potentially fatal complications, which may lead to an immediate risk of life. Thus, the detection and classification of arrhythmias is a pertinent issue for cardiac diagnosis. (1) Background: To capture these sporadic events, an electrocardiogram (ECG), a register containing the heart’s electrical function, is considered the gold standard. However, since ECG carries a vast amount of information, it becomes very complex and challenging to extract the relevant information from visual analysis. As a result, designing an efficient (automated) system to analyse the enormous quantity of data possessed by ECG is critical. (2) Method: This paper proposes a hybrid deep learning-based approach to automate the detection and classification process. This paper makes two-fold contributions. First, 1D ECG signals are translated into 2D Scalogram images to automate the noise filtering and feature extraction. Then, based on experimental evidence, by combining two learning models, namely 2D convolutional neural network (CNN) and the Long Short-Term Memory (LSTM) network, a hybrid model called 2D-CNN-LSTM is proposed. (3) Result: To evaluate the efficacy of the proposed 2D-CNN-LSTM approach, we conducted a rigorous experimental study using the widely adopted MIT–BIH arrhythmia database. The obtained results show that the proposed approach provides ≈98.7%, 99%, and 99% accuracy for Cardiac Arrhythmias (ARR), Congestive Heart Failure (CHF), and Normal Sinus Rhythm (NSR), respectively. Moreover, it provides an average sensitivity of the proposed model of 98.33% and a specificity value of 98.35%, for all three arrhythmias. (4) Conclusions: For the classification of arrhythmias, a robust approach has been introduced where 2D scalogram images of ECG signals are trained over the CNN-LSTM model. The results obtained are better as compared to the other existing techniques and will greatly reduce the amount of intervention required by doctors. For future work, the proposed method can be applied over some live ECG signals and Bi-LSTM can be applied instead of LSTM.

## 1. Introduction

Cardiovascular (CVD) diseases are globally recognised as the main cause of death, and they manifest themselves in the form of myocardial infarction or heart attack. According to the WHO [1], CVD is responsible for 17.7 million deaths. Approximately 31% of all deaths occur in poor and middle-income nations, with 75% of these deaths happening in these countries. Arrhythmias are the type of CVD that represents irregular patterns of heartbeats, such as the beating of the heart too fast or too slow. Examples of arrhythmias include: a trial Fibrillation (AF), premature ventricular contraction (PVC), ventricular fibrillation (VF), and Tachycardia. Although single cardiac arrhythmias may have little impact on one’s life, a persistent one might cause fatal problems, such as prolonged PVC that occasionally turns into Ventricular Tachycardia (VT), or Ventricular Fibrillation that can immediately lead to heart failure. Ventricular arrhythmias are one of the most prevalent types of cardiac arrhythmias that result in irregular heartbeats are responsible for nearly 80% of sudden cardiac deaths [2,3]. If arrhythmia conditions are detected early enough, ECG signal analysis may improve the identification of risk factors for cardiac arrest. Thus, it is fair to infer that monitoring heart rhythm regularly is crucial for avoiding CVDs. Practitioners use electrocardiographs as a diagnostic tool to detect cardiovascular diseases called arrhythmias, which detects and interprets the heart’s electrical activity during the diagnosis and is represented in ECG signals. However, ECG signals are represented in the form of waves when an ECG machine is attached to the human body; to get an exact picture of the heart, ten electrodes are needed for capturing 12 leads (signals). According to Zubair et al. [4], 12 ECG leads are required to properly diagnose, which are divided into precocious leads (I, II, III, aVL, aVR, aVF) to precordial leads ($V_{1}$, $V_{2}$, $V_{3}$, $V_{4}$, $V_{5}$, $V_{6}$). P waves, Q waves, R waves, S waves, T waves, and U waves based on the heart’s anatomy are illustrated in Figure 1. as positive and negative deflections from the baseline that signify a particular electrical event.

Conventionally, arrhythmia diagnosis studies focused on the filtering of noise from ECG signals [5], waveform segmentation [6], and manual feature extraction [7,8]. Various scientists have tried to classify arrhythmias using different machine learning (ML) and data mining methodologies. Here, we discuss some of the machine learning and deep learning techniques for the classification of arrhythmias.

Sahoo et al. [9] identified QRS complex using Discrete Wavelet Transform and Empirical Mode Distribution (EMP) for noise reduction, and Support Vector Machine was used to classify 5 distinct kinds of arrhythmias with an accuracy of 98.39% and 99.87% sensitivity with an error rate of 0.42. Osowski et al. [10] used higher order statistics (HOS) and Hermite coefficients to detect the QRS complex. The performance of the proposed approach [10] is compared with spectral power density algorithm, genetic algorithm, and SVM for classification of 5 different types of arrhythmia, which provides an average accuracy of 98.7%, 98.85%, 93%. Although these models are quite accurate due to the manual feature extraction process, their computation cost is high. Plawiak et al. [11] used higher-order spectra for feature extraction, PCA for dimension reduction, and SVM to identify 5 different forms of arrhythmia with the accuracy of 99.28%.

Yang et al. [12] used HOS and Hermite functions for manual feature extraction, and they also used SVM for disease classification. Polat and Gunes et al. [13] suggested least square SVM (LS-SVM) for the classification of arrhythmia and used PCA to reduce the dimensionality of features from 256 to 15. Melgani and Bazi et al. [14], using SVM, carried out an experimental study to identify five types of irregular waveforms and natural beats in a database from the MIT-BIH dataset. To increase the efficiency of SVM, they adopted particle swarm optimization (PSO). The sole purpose of PSO is to fine-tune the discriminator function for selecting the best features for SVM classifier training. The authors [14], contrasted their technique to two existing classifiers, namely K-nearest neighbours (KNN) and radial basis function neural networks, and discovered that SVM outscored them with an ≈90% accuracy. In a similar vein, to optimise the discriminator function, GA is used in [15]. Furthermore, they used a modified SVM classifier for arrhythmia classification.

Dutta et al. [16] suggested an LS-SVM classifier for heartbeat classification, such as normal beats, PVC beats, and other beats, by utilising the MIT-BIH arrhythmia database. Their approach yields 96.12% accuracy for classification. Desai et al. [17], considering 48 records of MIT-BIH arrhythmias, suggested a classifier using SVM for the classification and Discrete Wavelet Transformation (DWT) for feature representation. They categorised five types of arrhythmia beats: (i) Non-ectopic (N), (ii) Supraventricular ectopic (S), (iii) Ventricular ectopic (V), (iv) Fusion (F), and (v) Unknown (U). Their approach resulted in an accuracy of 98.49%. Furthermore, the authors performed statistical analysis using ANOVA to validate the efficacy of the proposed approach.

Kumaraswamy et al. [18], considering the MIT-BIH arrhythmia database, proposed a new classifier for the classification of heartbeats that is useful for the detection of arrhythmias. In particular, they used a random forest tree classifier and discrete cosine transform (DCT) for discovering R-R intervals (an essential pattern that helps in detecting arrhythmias) as features.

Park et al. [19] proposed a classifier for detecting 17 different types of heartbeats that can be used to detect arrhythmias. They carried out a two-step experimental study: (a) in the first step, P waves and the QRS complex are identified using the Pan-Tompkins method, and (b) in the second step, KNN classifier is used to classify them. This model was validated using MIT-BIH arrhythmias and performed with a sensitivity of 97.1% and specificity of 96.9%. Jun et al. [20] used a high-performance GPU-based cloud system for arrhythmia detection. Similar to [19], they used the Pan-Tompkins algorithm and KNN for identification and classification.

Machine learning paradigms are heavily influenced by feature architecture and a focus on feature extraction and filtering. The underlying concept behind learning is to include all of the data in signals so that the machine learning algorithm can learn and choose functions. This theory also underpins the deep learning paradigm, especially CNN and its 1-D equivalents [21]. Because of the potential and prospect of deep learning techniques, researchers [21,22,23,24,25] have adopted these techniques for the detection/classification of various types of chronic diseases. In this direction, Acharya et al. [22,23,26,27] conducted a detailed study for arrhythmia classification using deep learning. In [26], they proposed an automated classifier for arrhythmias. In [23], a CNN architecture is built to predict myocardial infarction with a 95.22% accuracy. In [27], an automated CNN model for the categorization of shockable and non-shockable ventricular arrhythmias is proposed. The suggested model outperformed with an accuracy of 93.18%, 95.32% sensitivity, and 94.04% specificity.

Kachuee et al. [24] utilized deep residual CNN for classification, and t-SNE was used for visualization. They employed their approach to identify five distinct arrhythmias by the AAMI EC57 norm and offered better results than ALTAN et al. [25] on the same database of MIT-BIH arrhythmias with an accuracy of 95.9%. Xia et al. [28] utilised short-term Fourier transform (STFT) and stationary wavelet transform (SWT) for the detection of arterial fibrillation arrhythmias. Further, to evaluate ECG segments, STFT and SWT outputs are input to two separate deep CNN models. As a result, CNN outperformed with 98.29% for STFT and 98.63% for SWT. Savalia et al. [29] proposed a model based on 5-layer CNN that can classify five types of arrhythmias. Although these models are quite accurate for minimising the loss function in backpropagation, they still suffer from the vanishing gradient problem of exploding gradients. For the classification of cardiac arrest during CPR, Jekova et al. [30] employed an end-to-end CNN model entitled CNN3-CC-ECG, which they validated on the independent database OHCA. With automatic feature extraction, the model performed well, with a sensitivity of 89% for ventricular fibrillation (VF), a specificity of 91.7% for non-shockable organised rhythms (OR), and a specificity of 91.11% for asystole, but found moderate results with noisy data. Krasteva et al. [31] applied convolution DNN for the classification of shockable and non-shockable rhythms, and to optimise the model they used random search optimization. Elola et al. [32] utilized two DNN architectures to classify pulseless electrical activity (PEA) and pulse-generating rhythm (PR) using short ECG reading segments (5s) on an independent database (OHCA). Additionally, the model was optimised using Bayesian Optimization. Both architectures work well with balanced accuracy of 93.5% as compared to other state-of-the art algorithms. Although, 1D CNN outperformed with automatic feature extraction [33] and more accurate than clinicians [34], but recurrent neural networks (RNNs) are effective deep learning methods as they can handle time dependencies and variable-length segmented data [35]. Although RNNs face certain difficulties, such as the vanishing gradient problem [36], in which the gradient decreases with backpropagation and its value becomes too small, Consequently, diminished gradient values do not contribute much to learning. The reason for this is that with RNN, the layers that receive a diminished gradient to upgrade weights stop learning; as a result, these layers do not remember and may not be able to recall the longer sequences used. Thus, these layers have a short-term memory, which impacts negatively predicted problems. Nevertheless, this vanishing gradient issue can be resolved by using LSTM or GRU with ReLU, which allows capturing the impact of the earliest available data. Actually, by tuning the burden value, the vanishing gradient problem can be avoided. Altan et al. [25] presented a classification network based on a four-layer deep belief network to classify five kinds of arrhythmias with an accuracy of 94%. Furthermore, they used DFT for feature extraction.

Ping et al. [37] have presented an 8CSL approach for the identification of atrial fibrillation that includes shortcut connections in CNN, which aids in boosting data transmission speed, and 1 layer of LSTM, which aids in reducing long-term dependencies between data. To further test the proposed methodology, he compared it to the RNN and the multi-scale convolutional neural network (MCNN), and he discovered that 8CSL extracted features better when compared to the other two methods in terms of F1-score (84.89%, 89.55%, 85.64%) with different data segment lengths (5 s, 10 s, 20 s). When it comes to heartbeat detections, Ullah et al. [38] used three different algorithms: CNN, CNN+LSTM and CNN+LSTM+attention model for the classification of five different types of arrhythmia in heartbeat detections over two well known databases, MIT-BIH arrhythmias and the PTB Diagnostic ECG Database. They discovered that CNN had 99.12%, CNN+LSTMhad 99.3%, and CNN+LSTM+attention model had 99.29%, all of which were statistically significant. Kang et al. [39] tries to classify mental stress data using the CNN-LSTM model, where he has trained his model using the ST Change Database and WESAD Database, and he has converted the 1D ECG signals into the time and frequency domain in order to train the proposed method. On testing, he achieved an accuracy of 98.3%.

The majority of past work has trained models using a 1D ECG output, which contains a lot of noise, such as baseline wandering effects, power line interference’s, electromyographic noises [40], and artefacts. Filtering and extracting features requires numerous preprocessing processes, which can jeopardise data integrity and model accuracy. Thus, the main aim of this study is to automate the detection and classification process. For that, 1-D ECG signals are translated into 2-D colored scalogram images using the Continuous Wavelet Transform (CWT) with a resolution of 227 × 227 × 3. Thereafter, a 2D CNN is adopted for automatic feature extraction and an LSTM is adopted for classification purposes. In sum, the following are the paper’s key contributions. (1) We proposed a hybrid approach combining the power of CNN and LSTM. In the initial phase, we converted all the ECG signals to images using CWT, so that the CNN model executes effectively, and then applied LSTM to classify arrhythmias. The approach was developed using the deep learning library on the MATLAB platform. The performance of the proposed approach is validated using the well-known MIT–BIH Arrhythmia database in two ways: firstly, by dividing the dataset into 75% for training and 25% for testing; and secondly, by training the model using K-fold cross validation. Additionally, the parameters of the proposed model are hyper-trained with the grid search algorithm.

Rest of the paper is organised in the following manner: The detailed description of the methodology, including MIT-BIH arrhythmias, dataset preprocessing steps to filter data and conversion of ECG signals into scalogram images, is given in Section 2. Experimental results and performance evaluation are reported in Section 3. Discussion is mentioned in Section 4; Finally, Section 5 concludes the paper.

## 2. Methodology

This section describes the dataset utilized, data cleaning and preprocessing techniques, and a thorough explanation of the suggested model.

### 2.1. Dataset

We evaluated the accuracy of our CNN-LSTM model using 162 ECG recordings from three Physionet databases (https://archive.physionet.org/physiobank/database, accessed on 1 February 2021). A description about these is as follows.


96 recordings: taken from the MIT-BIH cardiac arrhythmias database [41,42], this repository contains beat annotation files for 29 long-term ECG recordings of individuals with congestive heart failure ranging in age from 34 to 79. Eight men and two women were among the subjects, while the gender of the remaining 21 was uncertain, and the initial ECG recordings were digitised at 128 samples per second. The sampling frequency of MIT-BIH cardiac arrhythmia is 360 Hz with a resolution of 5 µV/bit.36 recordings: was taken from the MIT-BIH normal sinus rhythm database, which contains 18 long-term ECG recordings of patients hospitalised at Boston’s BIH arrhythmia laboratory. The patients in this database, which comprised five males aged 26 to 45 and thirteen women aged 20 to 50, had no severe arrhythmias. The MIT-BIH database for normal sinus rhythm is sampled at 128 Hz and the data is accessible at uniform intervals of 7.8125 ms. These signals are digitised using a 12-bit analog-to-digital converter (ADC) at a sampling rate of 128 Hz [41].30 recordings: taken from the BIDMC congestive heart failure database [41,43]. this repository includes 15 patients’ long-term ECG recordings with severe congestive heart failure (NYHA class 3–4) (11 men, ages 22 to 71, and 4 women, ages 54 to 63). The duration of each recording is about 20 hours, with two ECG signals sampled at 250 samples per second and 12-bit resolution over a ten-millivolt spectrum. At Boston’s Beth Israel Hospital, the first analog recordings were produced using ambulatory ECG recorders with a recording bandwidth of about 0.1 Hz to 40 Hz.


According to [22], the database is structured as an array of two fields: data and labels. Every recording consists of 65,536 samples. As a result, the data is interpreted as a 162 × 65,536 matrix, which means it comprises a total of 162 ECG signals with a sample size of 65,536 and is re-sampled at a standard rate of 128 hertz. In contrast, labels indicate ECG signal information, that is, ARR signals are from 1st to 96th row of an array, 97th to 126th rows of an array represent CHF signals. However, from 127th to 162th row represents NSR signals, as seen in Figure 2.

### 2.2. Data Preprocessing

This stage is used to prepare data for training and testing. First, the data is segmented by using a transformed data store and the resize data helper function. Continuous Wavelet Transformation (CWT) was also used to turn 1-D ECG signals into 2-D colored scalogram images.

#### 2.2.1. Data Segmentation

Deep Learning models, such as CNN, are dynamic models used for feature extraction and require a lot of data for training the process model. When exceptionally lengthy input signals are sent through the CNN network, the estimated performance may suffer due to the degradation. The ECG signals and their related label masks should be broken up using a modified datastore and the resizeData helper function to prevent these side effects. Our analysis used a dataset from the Physionet databases that included ECG recordings from 162 patients, and each patient recording consisted of 65,536 sequence segments. Here, we have divided 65,536 sequence segments into ten chunks of 500 samples each and discarded the remaining parts of the segment. In our study, we have taken 96 recordings from ARR, 30 recordings from CHF, and 36 recordings from NSR. To make them equally proportional, we used 30 recordings from the NSR database, 30 recordings from the CHF database, and 30 recordings from the ARR database. As a result, there are 900 recordings, and each recording is divided into ten chunks of 500 samples each, which is a lot of data to train CNN for feature extraction and LSTM for classification. Table 1 describes the information in all arrhythmia databases from Physionet.

**Remark** **1.**
*In our experiment study, we computed $floor*\frac{65,536}{500}=131$ chunks, out of which we chose only 10 chunks. The length of each chunk is 500, which passes through CWT 12 band pass filters and is converted into scalogram images, which represent the ECG signals in the time-frequency domain. Finally, we have 900 scalogram images to train the proposed model with K-fold cross validation.*


#### 2.2.2. Image Conversion

The majority of previous work has used a 1D ECG signal to train models, which has a lot of noise [44] such as baseline wandering effect [40], and artifacts. They need many preprocessing steps to filter and extract features, which can compromise data integrity and model accuracy. Thus, in this study, 1D ECG signals are transformed into 2D colored scalogram images using CWT, which are used as input parameters with a resolution of 227 × 227 × 3. As seen in Figure 3, where a normal sinus rhythm signal is converted into a scalogram image using CWT. Application of CWT is denoising and conversions of 1D signals into 2D scalogram images.

#### 2.2.3. Continuous Wavelet Transformation (CWT)

The CWT essentially allows the signals to be mapped into a time-scale domain. It also makes the frequency components in the studied signals more visible. Following previous studies [45,46], this study considers the CWT as a potential candidate solution. Before defining the CWT, let us describe the required machinery setup.

**Definition** **1**(Wavelet). *A wavelet is defined as a limited duration waveform, whose average value is zero. The wavelet function for frequency is defined as:*
(1)$$\varphi_{p,q}\left(t\right)=\frac{1}{\sqrt{p}}\varphi *\frac{\left(t-q\right)}{p},\;p,q\in R_{+}$$

**Definition** **2.**
*Continuous wavelet transformation (CWT) is the sum of the signal multiplied by shifted and scaled versions of the wavelet function φ. The components of functions φ(t) are called mother wavelets and $\varphi_{p,q}\left(t\right)$ are called daughter wavelets; and it is obtained by comparing signals by stretching and compressing of mother wavelet at various scales and positions. CWT [47] φ represents the usage of a wavelet to quantify the similarity between a signal and an analysis function. CWT is a technique that is often used to denoise and depict ECG data in the time and frequency domains. as scalogram pictures.*


We can obtain CWT coefficients of two variables Cp,q of the region R where *p* is used for scaling purpose and *q* is used for positioning on *x* axis, and ∗ denotes the complex conjugate, and *t* represents the time interval. At lower scales, frequencies are high because waves are less stretched, but as we move to higher scales, the frequency of waves decreases as waves are more stretched at this position. By analyzing the coefficients, more variations can be seen at lowers scales where the frequency is very high, so one can able to capture more varying details of signal f(t), and at higher scales where the waves are less stretched, a minor variation of coefficients can be seen and able to capture less varying details of the signal. The diagrammatical demonstration for CWT coefficient computation is shown in Figure 4.

**Remark** **2.**
*In our study, we used 2D scalogram images as input for training and validation of our model, and CWT was used to convert 1D ECG signals into 2D scalograms. We have used Morlet wavelet (i.e., amor) which has a one–sided spectra and has complex values in the time domain. Twelve bandpass filters per octave are used for CWT, as shown in Figure 5.*


### 2.3. Model Architecture and Details

There are several methods to automatically analyse ECG, including machine learning-based [48,49] and deep learning-based [34,50,51]. Deep learning algorithms are more feasible because feature engineering is performed automatically (e.g., extraction of QRS Complex). Here, the mapping of ECG input signals into types of arrhythmias is performed end-to-end. This section explains the complete architectural details to identify three types of arrhythmias (i.e., ARR, CHF, and NSR). First, 1D signals are segmented using Section 2.2.1, and the 2D scalogram images are made using CWT, which helps convert the 1D signals into the time and frequency domain. Then these scalogram images are classified using a structure that is a combination of 2D-CNN and LSTM. The proposed model is trained with K-fold cross-validation, and hyper-parameter tuning is performed through a grid search optimization algorithm. As shown in Figure 6.

#### 2.3.1. Convolutional Neural Network

Convolutional Neural Network (CNN) [52] is a neural network used for enhancing the image or fetching some helpful information out of it, for example, image classification, and by taking 2-D grid features of an image and finding temporal features of time series data by taking 1-D grid samples at different time intervals. The basic architecture and its characteristics are described in [10,53].

The main functions of CNN are max pooling, convolution, classification, and non-linearity. In this analysis, CNN is in charge of extracting temporal features, while LSTM excels at capturing the helpful parameters of time series data and classification. Now, the layers of CNN are in order.


2D convolution layer: is in charge of generating feature maps from 2-D filters strung together.ReLU Activation Function: Activation is an incredibly critical function of a neural network. The responsibility of an activation function is to determine whether the information revived by a neuron is useful or can be overlooked. In this study, we used ReLU as an activation mechanism. It is a non-linear activation function that is used to reduce the linearity of an image by deactivating neurons only when their values are less than zero (explained through (4)).
(2)$$Y=ReLU\left(\sum (W*X)+b\right),$$
where, *X* are inputs, *W* represents weights and *b* represents bias and *Y* represents output value.Batch normalization: The parameters in the previous layer may significantly impact the input distribution of the further layer. Batch normalisation is the most important layer since it normalises the last layer’s output and acts as a regularizer to deter the algorithm from overfitting. This effect is referred to as “internal covariate transition”. Batch normalisation estimates the mean and variance of input batches and normalizes, scales, and shifts them. In this work, batch normalisation is performed after the activation. The formula for batch normalisation is calculated as:
(3)$$\alpha =\frac{1}{n}\sum_{i=1}^{n}y^{i},$$
(4)$$\beta^{2}=\frac{1}{n}\sum_{i=1}^{n}y^{i}-\alpha ,$$
(5)$$y^{\left(i\right)}=\frac{y^{\left(i\right)}-\alpha }{\sqrt{\beta^{2}+\epsilon }},$$
where: α = mean of same batch; β = variance of same batch; $y^{\left(i\right)}$ = standardized output value; ϵ = constant value as 0.001.Max pooling: is used for dimensionality reduction or downsampling of input matrices. Max pooling is achieved by adding the maximum filter to non-overlapping sub-regions and selecting the maximum value from each patch.Flattening: a 2-D matrix is transformed into a 1-D vector that can be fed into a completely connected layer by this layerSoftmax layer: is used to measure the probability distribution of an occurrence over *n* separate events in a multi-class grouping. This feature computes the probabilities for each target class in the total classes, ranging from 0 to 1. The probabilities are then used to decide which target class has a higher probability for the given inputs.Classification layer: is used for classifying different categories.Dropout Regularization: generally, the network suffers from over-fitting during the model training [51]. Dropout regularisation solves this problem by discarding some of the function nodes and reducing dependencies between them. We used a 50% dropout before the last completely linked layer in our model.


#### 2.3.2. Long Short Term Memory (LSTM)

Traditional artificial neural networks have limitations in that they are unable to collect the sequential information required for dealing with sequence data in the input data [54,55]. However, in making predictions, RNNs extract sequential information from the input data, that is, the connection between the words in the text. The RNN’s future hidden state is estimated as follows: Let at given time stamp vector t=(1,⋯,T), input $x=(x_{1},\cdots ,x_{T})$, the output $q=(q_{1},\cdots ,q_{T})$, future hidden state vector $h=(h_{1},\cdots ,h_{T})$ and mathematically determined as:(6)$$h_{t}=H\left(W_{xh}x_{t}+W_{hh}h_{t-1}+b_{h}\right),$$
where: $q_{t}=W_{hq}h_{t}+bq$, W: weight matrices, $W_{xh}$: weight matrix between input and hidden vector, $b_{h}$: the bias vector for hidden state vectors, H: hidden layer activation function.

The staple concern in conventional RNN is that the back-propagation step is to attenuate the loss function gradient, and its value becomes so tiny that it does not contribute much to learning. As these layers get a slight gradient to upgrade their weights, they stop learning, and this is called the vanishing gradient problem [56]. Thus, they have a short-term memory that has detrimental effects on prediction problems by restricting the network for training in long-term dependencies. Knowing these limitations, we resort to the LSTM that uses cell state memory rather than simple neurons, as shown in Figure 7.

LSTM architecture [57]: consist of following components: Forget Gate, Input Gate and Output Gate. These gates can display or erase information in the cell state memory over random time intervals explained through the following functions [58].

Forget Gate: it allows or discard information. Forget vector at timestamp *t* is calculated as:(7)$$F_{t}=\sigma \left(W_{xf}x_{t}+W_{hf}h_{t-1}+W_{cf}C_{t-1}+b_{f}\right),$$
where $W_{xf}$ is the weight vector between input and forget gate; $x_{t}$ is the current input value at timestamp *t*; $W_{hf}$ is the weight vector between hidden state and forget gate; $h_{t-1}$ is the previous hidden state at time t−1; $W_{cf}$ is the weight vector between cell state vector and forget vector; $C_{t-1}$ is the previous cell state at $C_{t-1}$; $b_{f}$ is the bias vector for setting weights for forget gate. When the summation of all these values is passed through the sigmoid function, if its values are within the range of 0 and 1, the gate allows it to pass; otherwise, it simply discards its information.

Input Gate: Transfer current values and previous values to the sigmoid activation function, which allows you to refresh the cell state memory only when the values are between 0 and 1. Input vector at timestamp *t* is calculated as:(8)$$i_{t}=\sigma \left(W_{xi}x_{t}+W_{hi}h_{t-1}+W_{ci}C_{t-1}+b_{i}\right),$$
where $W_{xi}$ is the weight vector of input values; $W_{hi}$ is a weight vector between input gate and current values; $h_{t-1}$ previous hidden state at time t−1; $W_{ci}$ is the weight vector between input gate and cell state memory.

Cell State: Sets the current cell state, multiplies the Forget variable with the previous cell state and drop values if multiplied by almost 0, applies the output of the input gate to alter the value of the cell state. Cell state vector at timestamp *t* is calculated as:(9)$$C_{t}=F_{t}C_{t-1}+i_{t}\tanh\left(W_{xc}x_{t}+W_{hc}h_{t-1}+b_{c}\right).$$
where, $W_{xc}$ is the weight vector between the input vector and the cell state vector; $W_{hc}$ is the weight vector between the current hidden state and the cell state state vector; $b_{c}$ is the bias vector of the current cell state.

Output Gate: For the sake of prediction, it determines what the next hidden state is. Output vector at timestamp *t* is calculated as:(10)$$o_{t}=\sigma \left(W_{xo}x_{t}+W_{ho}h_{t-1}+W_{co}C_{t-1}+b_{o}\right).$$

Finally, the next hidden state value is calculated by applying the hyperbolic activation function to the current cell state memory and by doing a dot product with the output gate vector:(11)$$h_{t}=o_{t}\cdot \tanh\left(c_{t}\right)$$

**Remark** **3.**
*Sigmoid activation function, σ(·), is a logistic function its value ranges from 0−1. It is mainly used for binary classification, formula for sigmoid function: $\sigma =1/(1+e^{-z})$. However, tanhis a hyperbolic activation function whose value ranges from −1 to 1, which provides the probabilities distribution for the input vector for multi-classification.*


#### 2.3.3. Classification of Arrhythmias Using Proposed (2D-CNN-LSTM) Model

In this section, we present the complete work for the classification of our proposed model 2D-CNN-LSTM through the 20 layers presented in Table 2 and visualised through Figure 8. Here we feed ECG scalogram image sequence data of size 227×227×3 into sequence input layer. Subsequently, these scalogram images are converted into an array form (vertical, horizontal, channel) using a sequence folding layer. Then these input features are passed through the first convolutional 2-D, which is responsible for creating a feature map of size 227×227×64, by using Equation (Equation 12), where we have applied 64 filters of size 3×3 along with padding and stride as 1, we will get the values of the output data.
(12)$$\left(OUT_{H},OUT_{W}\right)=\left(\frac{h+2p-fh}{s}+1,\frac{w+2p-fw}{s}+1\right)$$
where: $OUT_{H}$ = output height; $OUT_{W}$ = output weight; h,w = input data size; *p* = padding size; fh = height of filter; *s* = no of stride; fw = weight of filter.

Additionally, we introduce non-linearity to the output data using the ReLU function through Equation (Equation 2), which has a value in the range of 0-1. It deactivates neurons whose values are less than zero. Further, output data is passed through cross channel normalisation with five channels per element that regularises the meaning and prevents over-fitting of functions.

Next hidden state performs Max, Pooling which is used for the downsampling of the function diagram uses matrix of size [2×2] with stride 2 and padding [0000], as a result, the matrix size id reduced to 64×64×64. The effects of the downsampling of the feature diagram are transferred through many convolutional and pooling layers, and the matrix is transformed into 32×32×192. The input features are then flattened and encoded as 192×192 1D vectors, which are fed into the LSTM at layer 11. The flattening layer has the benefit of not altering the parameters by transforming the retrieved feature map output into a 1D array, allowing the feature maps to be rebuilt as input to the LSTM. At this point, the input of size 800×4096 is routed via the LSTM’s hidden layer. A weight value of 800 × 4096 is applied to Equations (7)–(10) to extract the feature value, which reflects the LSTM layer’s computational process. In the LSTM layer, there are three types of gates: an input gate ($i_{t}$), a forget gate ($f_{t}$), an output gate ($O_{t}$), and a cell state memory. Using the sigmoid and tanh functions, a weight value is multiplied by an input vector ($x_{t}$), a hidden state ($h_{t}-1$), and a cell state ($C_{t}$), and then a feature value is retrieved from the result of the multiplication. Subsequently, the feature value computed at the output gate is then sent to the output layer using Equation (Equation 11). The procedure of obtaining a needed feature value from multiple feature values generated at the output gate is described in Equation (Equation 11). The feature value in the range computed using the output gate is transmitted to the output layer after extracting a feature value from −1 to 1 using the Tanh function. Furthermore, these extracted features are passed through 3 fully connected layers used for classification, where we have applied a 50% dropout. A dropout is a regularisation approach that reduces overfitting in neural networks by removing some random nodes during the training process and improving generalisation error. These values are transferred into a classification layer that functions similar to ANN. Finally, it went into the Softmax activation function responsible for the multi-classification that measures the probability distribution of events over *n* of different events. This function measures the probability of each target class out of the whole target classes, and this likelihood ranges from 0 to 1. These probabilities help decide the target class with a greater likelihood and classify three different types of arrhythmias, such as ARR, NSR, and CHF. Components of the convolution 2D layer and LSTM layer are shown in Figure 9. Equation (Equation 12) depict the convolution 2D layer computation process in CNN models, whereas Equations (7)–(10) depict the process of outputting feature values in the LSTM layer utilising the weight values of the input gate, forget gate, and output gate. The feature values in the range are transferred from the output gate to the output layer using Equation (Equation 11).

Hyper Parameter Optimization: Tuning, or hyperparameter optimization, is a term used to describe the process of selecting a set of ideal parameters for machine learning algorithms. Hyperparameters are variables that may influence how a machine learns. For generalising data patterns, similar machine learning algorithms may have different learning rates, weights, and restrictions. Hyperparameters must be fine-tuned to optimally solve the issue. Finding a tuple that offers an ideal model and minimises the loss function is part of the optimization process. Hyperparameter tweaking may be performed in a variety of ways. We used the grid search approach for our research, which entails exhaustive searching of a portion of the algorithm’s hyperparameter space, followed by a performance measure. Data is scanned in the grid searching process to find the best parameters for a specific model. Parameters vary depending on the sort of model under consideration. Grid searching is not limited to a single kind of model; it may be used across the board in machine learning to find the optimum parameters for a given model. Grid searching creates a model based on every potential parameter combination, resulting in numerous iterations. Grid searching is computationally expensive since these model possibilities for each parameter are recorded. Grid search and random search are two typical strategies for tweaking hyperparameters. Every possible combination of hyperparameter values from a predefined list is tested in a grid search, with the best combination being selected based on the cross-validation score. Random combinations are chosen to train the model in a random search, allowing the number of parameters to be regulated. Although it effectively evaluates an extensive range of values and may quickly arrive at a solid combination, it has one fundamental flaw: it cannot guarantee the optimal parameter combination. Grid search, on the other hand, takes a long time yet produces the most significant results.

We employed grid search algorithms to obtain the top three mean test scores, such as 92.35, 83.63, and 78.5, that assist in achieving the maximum accuracy in real-time qualitative assessment, as shown in Table 3. We were able to get the optimal hyperparameters for training models by utilizing the maximum test score of 92.35, which included a learning rate of 0.01, epochs of 20, batch-size of 10, kernel-size of 1, hidden-units of 32, regularisation dropout of 0.05, and optimizer as Adam.

Cost Function: The cost function describes the difference between the given testing sample and the predicted performance and measures how well the neural network is equipped. The cost function is reduced using the optimizer function. Deep learning typically uses a cross-entropy function in several shapes and sizes. Mathematically, the cost function U is defined as:(13)$$U=\frac{1}{m}\sum \left(r\log(s)+(1-r)\log(1-s)\right),$$
where: *m* = batch-size; *s* = expected value; *r* = resultant value.

Using a gradient descent-based optimizer function with a learning rate, the cost function is minimized. Adam [60], Adagrad [61], and Adadelta [62] are some well-known optimizer functions. We found through experiments that the optimum point is quickly achieved when we use Adam. Therefore, we used the Adam optimizer algorithm, which had a learning rate of 1 × 10^−4^ and a decay rate of 0.95 for 1000 steps.

## 3. Experimental Results and Performance Evaluation

In this study, experiments were carried out on international standard ECG databases such as MIT-BIH cardiac arrhythmia (ARR), BIDMC congestive heart failure (CHF), and MIT-BIH normal sinus rhythm (NSR), which have detailed expert notation that is frequently utilised in contemporary ECG research. In this experiment, the dataset is divided using K-fold cross validation. Experimental tests were carried out on the workstation with an Intel Xeon gold 5215 10c 2.50 GHz processor and 16GB with training options: (Referenced from Table 3) MiniBatchSize as 10, MaxEpochs as 20, InitialLearnRate as 1 × 10^−1^, Learn Rate drop period as 3, Gradient Threshold as 1 and the total elapsed time for the training of the model is 6 min 30 s. These parameters are utilised using a grid search hyperparameter optimization algorithm with the highest mean test score of 92.35. Here, segmented 1D ECG signals were transformed into 2D scalogram images of size 227×227×3, and trained over the 2DCNN-LSTM model. Experiments were carried out on Matlab 2021a.

To do a qualitative evaluation of the proposed 2D-CNN-LSTM, we utilise six metrics [63] which are described as follows.


Sensitivity: determines the capacity of the model to accurately detect the actuality of the cases studied.Specificity: used to determine the ability of the model to distinguish between individual negative samples.Precision: defines the number of patients accurately classified by the model.Recall: quantifies the number of positive class projections made from all positive cases in the data setAccuracy: defines the percentage of correct classifications.F-measure: offers a single score that combines both the accuracy and recall.


The mathematical formulas for the above-mentioned metrics are given in Table 4, where True Positive (*TP*) represents the number of positive patients who have been assessed as positive for ARR. The True Negative (*TN*) represents the number of patients who are anticipated to be negative for ARR. Both of these matrices (*TP* and *TN*) indicate that the classification is accurate. A False Positive (*FP*) represents the percentage of patients who are categorised as positive but are negative for ARR. A False Negative (*FN*) is used to calculate the percentage of positive patients observed to be negative for ARR. However, both of the matrices (*FP*, *FN*) indicate a classification error. All three matrices (accuracy, sensitivity, and specificity) indicate the overall classification of the network model. The larger the value, the better the classification results will be.

Here, in this study, in order to evaluate the proposed model, we conducted two experiments. Firstly, the dataset was divided into two portions, 70% for the training dataset and 30% for the testing dataset, and by utilising hyperparameters (Referenced in Table 3) such as kernel size as [3 3]; number of filters as 64; batch-size as 10; regularization dropout as 0.05; optimizer as Adam, maximum pool Size as [2 2]; loss-method as binary cross entropy; InitialLearnRate as 1 × 10^−1^; decay as 0.0; epochs as 20 with 32 hidden units and found CNN-LSTM classifies three types of arrhythmias with an accuracy of ARR of 98%, precision of 0.97, recall of 0.96, F1-score of 0.98, sensitivity of 0.96, and specificity of 1, and for CHF class, accuracy of 77%, precision of 0.5, recall of 1, F1-score of 0.66, sensitivity of 1, and specificity of 0.69, and for the NSR class accuracy of 99%, precision of 0.98, recall of 1, F1-score of 0.66, sensitivity of 1, and specificity of 0.98, as shown in Figure 10 and referenced in Table 5. Training accuracy and training loss of the proposed model without K-fold validation are shown in Figure 11. Although the suggested models’ accuracy is acceptable, However, the model has under-fitting and over-fitting issues, which appear when the model has learnt less than or more than 20 epochs. The over-fitting issue model has a tendency to remember data and is unable to generalise new data, while the under-fitting model has a difficult time testing but is capable of generalising new data. We trained our models at 20 epochs and over-optimized parameters using 10-fold cross-validation to eliminate over-fitting and under-fitting, and the accuracy of proposed model was raised, (Referenced from Table 6) where the accuracy of ARR is 98.7%, precision is 1, recall is 0.98, F1-score is 0.98, sensitivity is 0.98, and specificity is 0.98, and for CHF class accuracy is 99%, precision is 0.97, recall is 0.96, F1-score is 0.96, sensitivity is 0.96, and specificity is 0.99 and for the NSR class accuracy is 99%, precision is 0.97, recall is 1, F1-score is 0.98, sensitivity is 0.97, and specificity is 0.99 as shown in Figure 12.

## 4. Discussion

In this study, an optimised ensemble model was developed using a combination of 2DCNN, which is used for automatic feature extractions, and LSTM, which has additional cell state memory and uses its previous information to predict new data. The goal was to improve the performance of arrhythmia classification while also avoiding overfitting. An optimization approach known as “grid search” was employed to tweak the model’s hyperparameters in order to optimise it. When compared to the random search method, it provides the best hyperparameters, despite the fact that it is computationally expensive. Additional K-fold cross validations were performed on international standard ECG databases such as the MIT-BIH cardiac arrhythmia (ARR), BIDMC congestive heart failure (CHF), and MIT-BIH normal sinus rhythm (NSR) in order to train the model. These databases contain detailed expert notation that is frequently used in contemporary ECG research. As part of our research, we employed the continuous wavelet transform (CWT), which is responsible for turning 1D ECG data into 2D scalogram images of size 227 × 227 × 3, which reflect signals in the time and frequency domain, and which we used for both training and validating the model. A confusion matrix, and other performance metrics were used to assess the classifier’s performance after it was enhanced using layers such as batch normalisation, a flattening layer, and a fully connected layer.

Further, we evaluated two distinct experiments with k-fold cross validation in the presence and absence of dropout regularisation. In Scheme A, dropout regularisation was not applied during the training process. Here, all weights were utilised in the learning process. However, in Scheme B, we applied dropout regularisation with 0.5 dropouts; as a result, 50% of the information was discarded, and only 50% of data is retained for learning. Results of both experimental schemes are visualized through Figure 13.

In the absence of dropout regularization, the results obtained from scheme A indicate high classification due to the overfitting of weights during training. The average accuracy is 99.8%, the average sensitivity is 99.77%, and the average specificity is 99.78%. However, in our suggested model where we have applied 0.5 of dropout regularization, only 50% of information is retained for learning. Hence, our suggested model has an average validation accuracy of 99%, average sensitivity as 99.33%, and average specificity as 98.35%, respectively, (Referenced from Table 7). However, validation accuracy for the classification of ARR is 98.7%, validation accuracy for CHF is 99%, and validation accuracy for NSR is 99% while sensitivity and specificity for ARR is 0.98%, 0.98% while sensitivity and specificity for CHF is 0.96% and 0.99%, and for NSR sensitivity and specificity is 0.97% and 0.99% respectively, (Referenced form Table 6).

Figure 14 represents the training progress and validation accuracy, as well as the training progress and validation loss. As seen in the graphs, training and validation errors were established at a value close to zero after 100 epochs, while training and validation accuracy stabilised at 98.76%, as seen in the graphs. These outcomes are very promising and had a high degree of precision (Referenced from Table 6).

The confusion matrix is derived from the training of a proposed model for classifying three types of arrhythmias. As evidenced by the confusion matrix, the model performs better for the categorization of CHF and NSR than ARR, which is average. This may be due to the slight morphological differences in the waveforms generated during the learning process. The testing dataset’s generated confusion matrix shows 99% accuracy for the normal sinus rhythm, 98.7% for cardiac arrhythmia, and 99% for congestive heart failure, as shown in Figure 12.

To validate our proposed methodology, we compared our results with the current and standard methods in terms of feature extraction methods, methodology, accuracy, and other statistical classifications as presented in Table 8. It is worth mentioning that the difference between the proposed work and the state-of-the-art described is rather promising in terms of accuracy and computing cost compared to other models, as seen in Figure 15. Knowing the potential and prospects of the proposed methodology, it would be interesting to use our proposed approach for diagnosing different critical diseases, such as gastrointestinal diseases, and distinguishing between neoplastic and non-neoplastic tissues.

## 5. Conclusions

Arrhythmia classification is the most crucial subject in healthcare. An arrhythmia is a rhythm or heart rate irregularity. This paper proposed an approach for the automated study of cardiac arrhythmias using the 2D-CNN-LSTM model. This approach has the following salient features as compared to conventional methods: (i) It employs the CWT to transform 1D ECG signals into 2D Scalogram colored images, making them ideal inputs for this network; (ii) It passes data from a 2D-CNN-LSTM, CNN for feature engineering and LSTM for classification, and has been trained on large labeled images.

Experiments on three ECG cross-databases (obtained under a range of acquisition conditions) showed their usefulness and ability to outperform other approaches in terms of classification performance. The confusion matrix for “normal sinus rhythm,” “cardiac arrhythmias,” and “congestive heart attacks” in the testing dataset showed 99% validation accuracy for “normal sinus rhythm,” 98.7% validation accuracy for “cardiac arrhythmias,” and 99% and validation accuracy for “congestive heart attacks.” Furthermore, sensitivity and specificity for ARR is 0.98%, 0.98% while sensitivity and specificity for CHF are 0.96 and 0.99%, and for NSR, sensitivity and specificity is 0.97% and 0.99% respectively. Likewise, the suggested model will assist clinicians in correctly identifying arrhythmias during routine ECG monitoring. According to preliminary results from the MIT-BIH database, our methodology’s overall efficiency is better than other methods.

Furthermore, the heavy computing burden caused by the use of CWT is a drawback. We could never achieve a complete inter-subject state, even though doing so will significantly minimise the amount of intervention required by doctors. It will be an excellent future avenue for researchers. A robust arrhythmia classification algorithm is needed to address these issues.

## 6. Further Details

In this section, we present the different functions used in the training of various deep learning models.

ReLU: Activation is an incredibly critical function of a neural network. These activation functions determine if the information neuron receiving is sufficient for the information or can be overlooked. We use ReLU as an activation mechanism in this study.

Batch Normalization: is the most critical layer used to normalize the performance of the previous layer and is often used as a regularization to prevent overfitting the model.

Softmax Function: is used to quantify the probability distribution of occurrences over *n* distinct events in multi-class classification. This function measures the probability of each target class of the whole target class, and this likelihood range from 0–1. Later, the probabilities help decide the target class with a greater likelihood for the target class.

Sigmoid Activation Function: is a logistic function its value ranges from 0–1 it is mainly used for Binary Classification.

Tanh Activation Function: It is similar to sigmoid activation function and its values ranges from −1,1.

Dropout: is used for regularization, which is used for controlling overfitting problems.

Learning Rate: its value ranges from 0–1 it defines how fast the model adapts the problem and how many weights are modified in the loss gradient model.

Hidden units: are the no of perceptrons used in neural network its values entirely lies on activation function.

Bias: modifies the activation function by applying a constant (i.e., a given bias) to the input. It is analogous to the position of constant in a linear function.

## Figures and Tables

**Figure 1 bioengineering-09-00152-f001:**
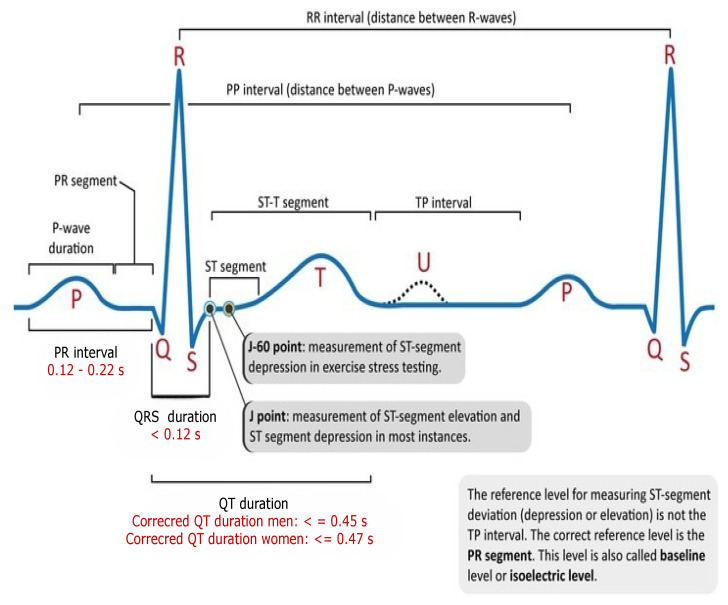
Representation of complete electrical activity of the heart.

**Figure 2 bioengineering-09-00152-f002:**
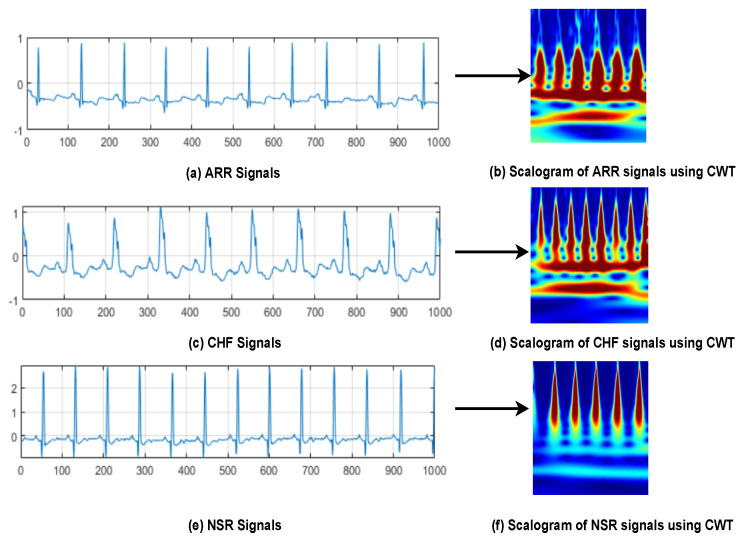
Distribution of ECG waveforms from three classes of arrhythmias. Sub-figures (**a**,**b**) Represents ARR signals and Scalogram image of ARR signal using CWT; Sub-figures (**c**,**d**) Represents CHF signals and Scalogram image of CHF signal using CWT; Sub-figures (**e**,**f**) Represents NSR signals and Scalogram image of NSR signal using CWT.

**Figure 3 bioengineering-09-00152-f003:**
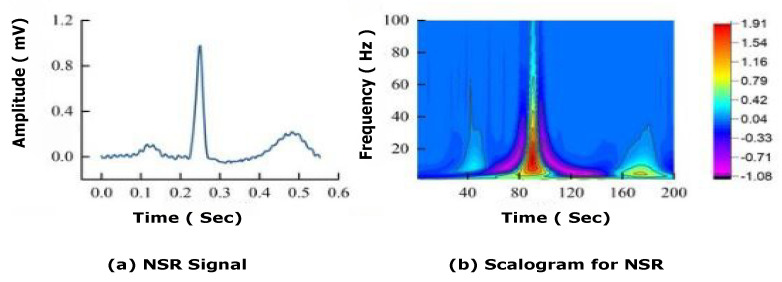
Conversion of normal sinus rhythm (NSR) signal into scalogram image. Sub-figures (**a**,**b**) represents NSR signals and CWT scalogram for NSR.

**Figure 4 bioengineering-09-00152-f004:**
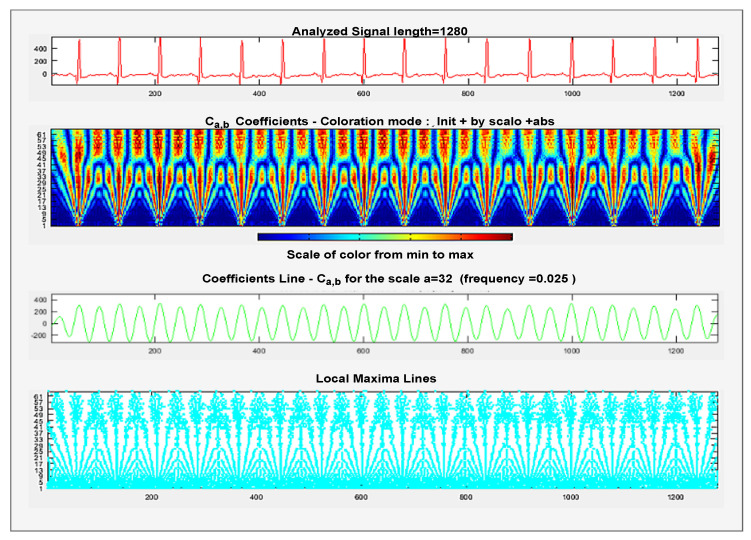
Different coefficient representation of Scalogram.

**Figure 5 bioengineering-09-00152-f005:**
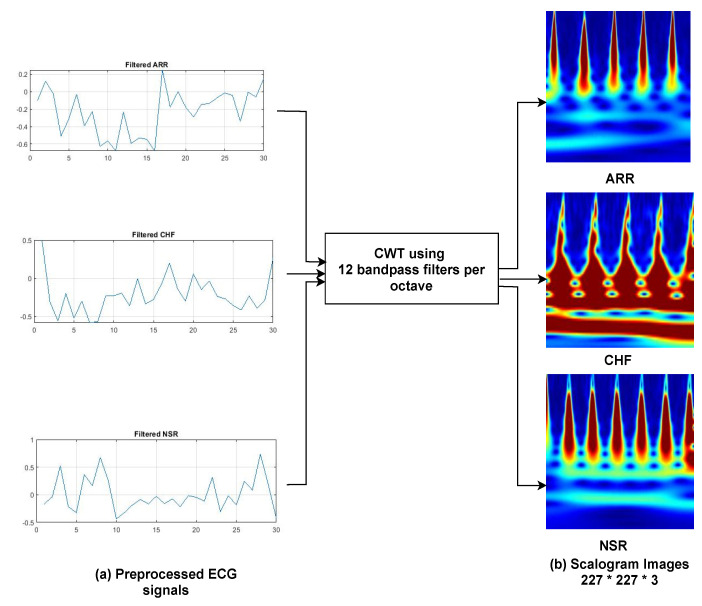
Conversions of 1D ECG signals into Scalogram images using CWT.

**Figure 6 bioengineering-09-00152-f006:**
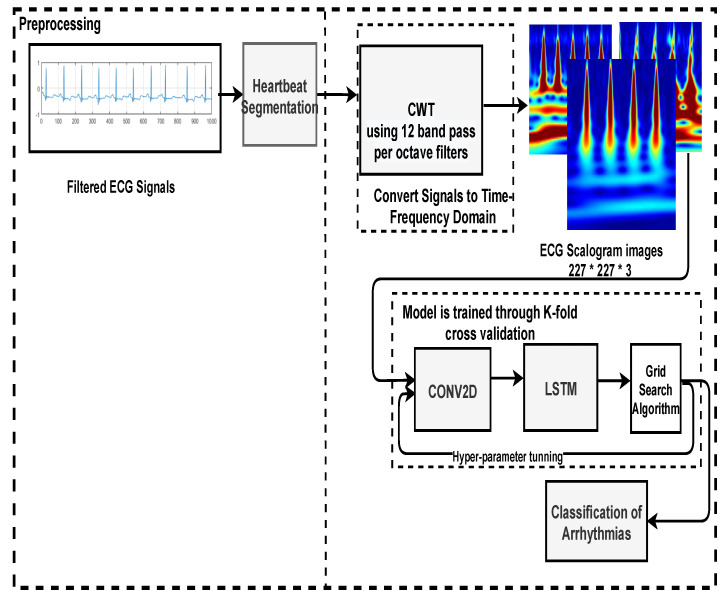
Block-diagram of the proposed method.

**Figure 7 bioengineering-09-00152-f007:**
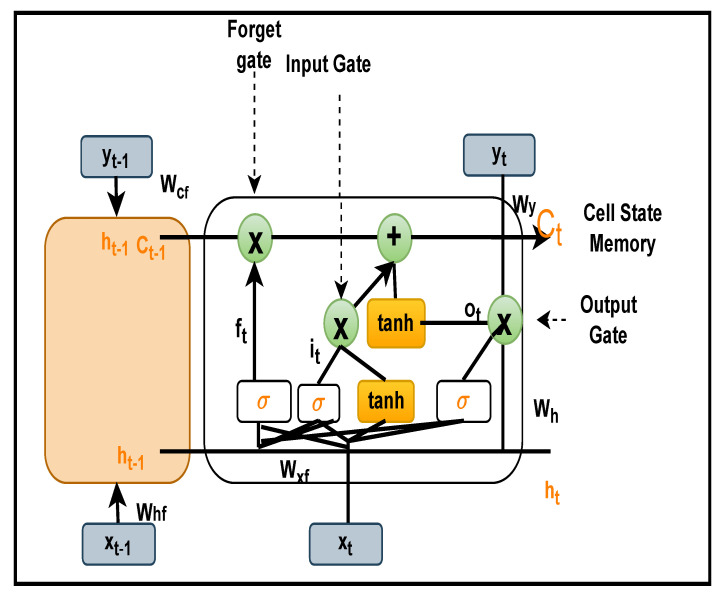
Complete representation of Long Short Term Memory Cell.

**Figure 8 bioengineering-09-00152-f008:**
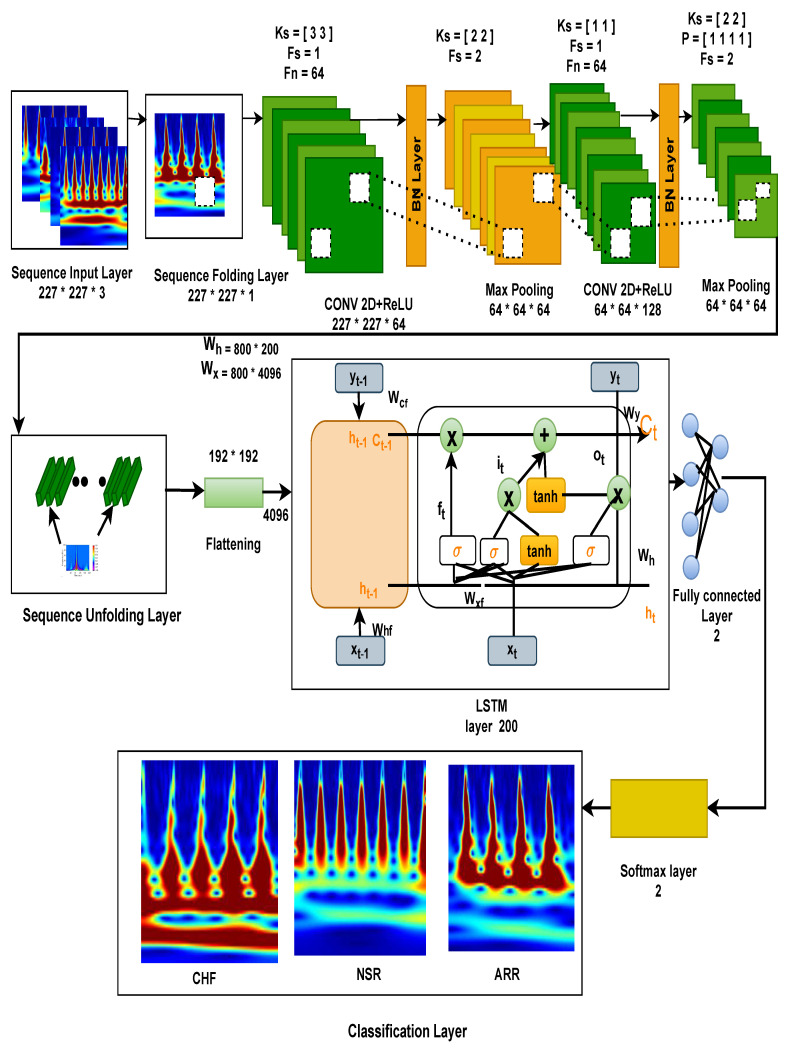
Representation of the internal architectural details of the proposed model.

**Figure 9 bioengineering-09-00152-f009:**
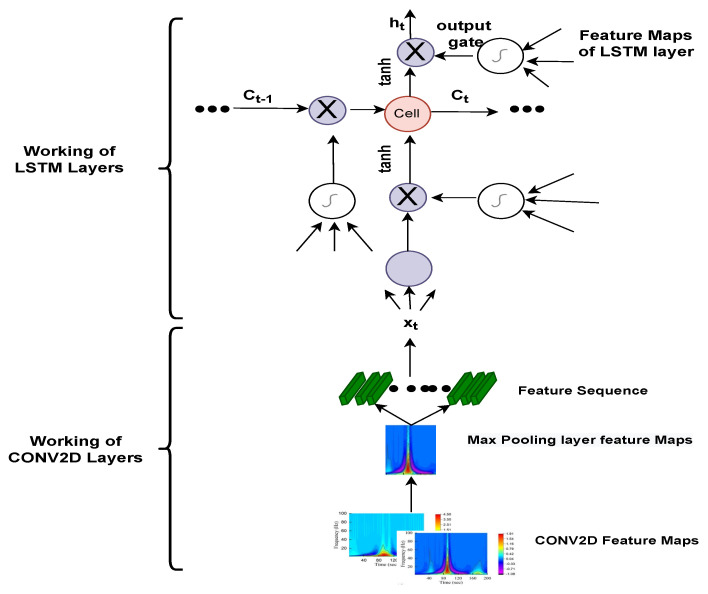
Internal components of the proposed 2D-CNN-LSTM model.

**Figure 10 bioengineering-09-00152-f010:**
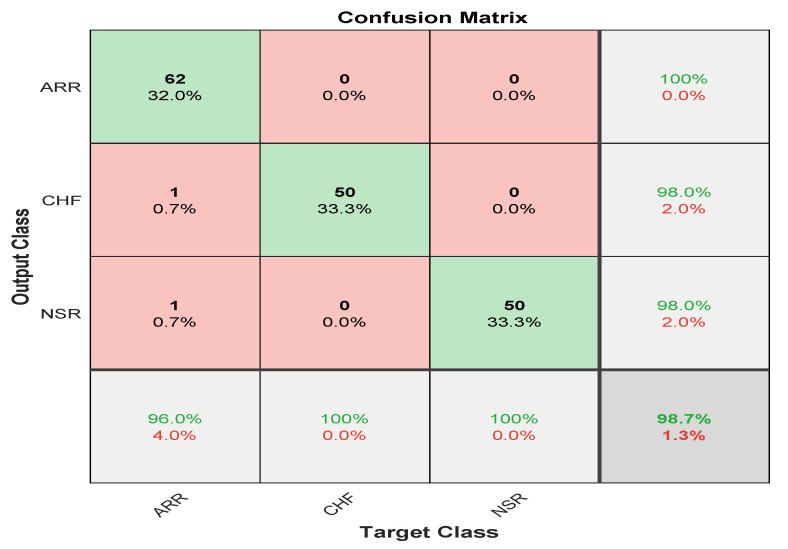
Confusion Matrix represents accuracy of the proposed model without k-fold cross validation 2D-CNN-LSTM.

**Figure 11 bioengineering-09-00152-f011:**
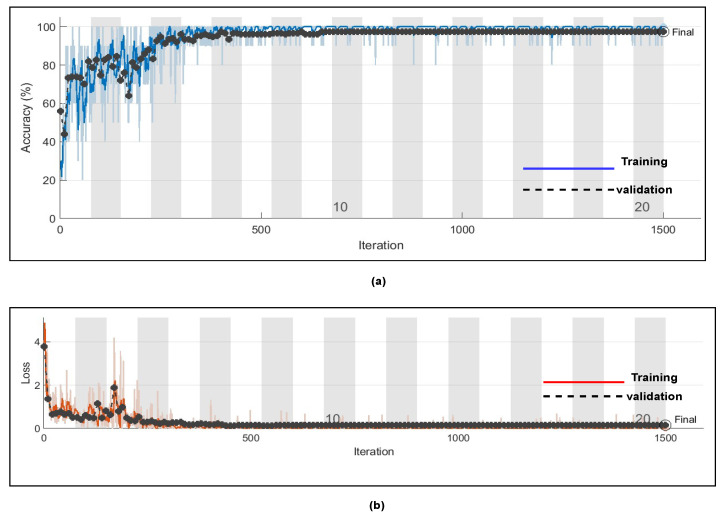
(**a**) Training progress and validation accuracy of proposed model without K-fold cross validation (**b**) training and progress loss of proposed model without K-fold cross validation.

**Figure 12 bioengineering-09-00152-f012:**
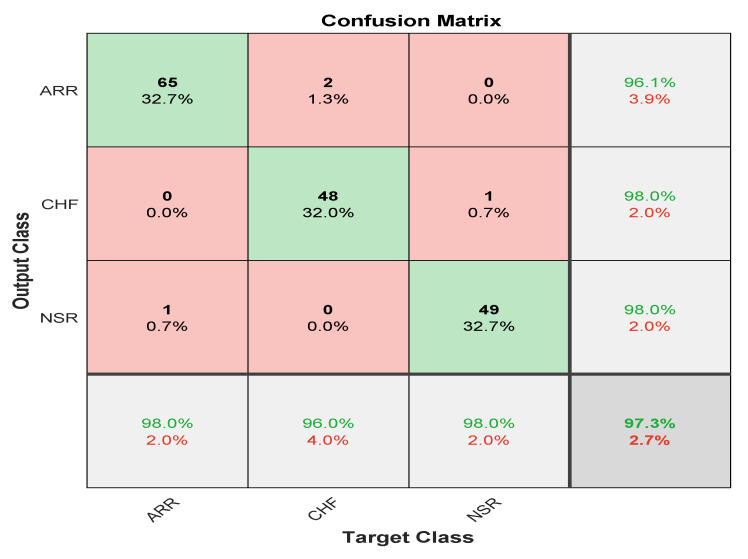
Confusion Matrix represents validation accuracy of proposed model with k-fold cross validation 2D-CNN-LSTM.

**Figure 13 bioengineering-09-00152-f013:**
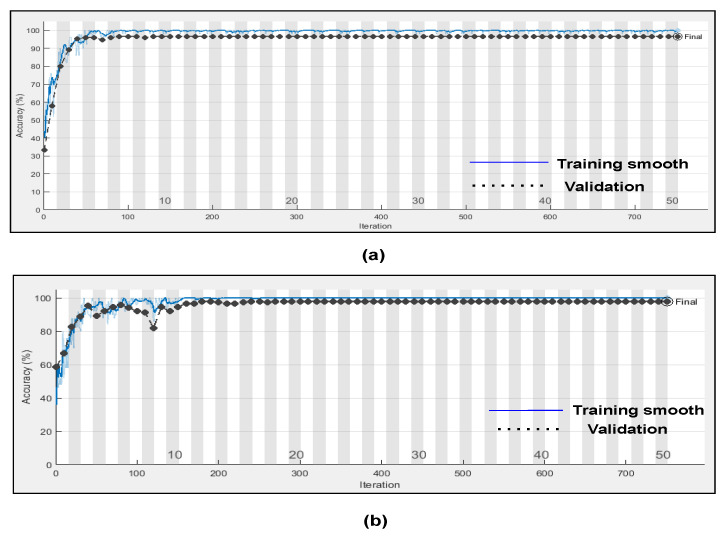
(**a**) Training progress and validation accuracy of proposed model with dropout regularization (**b**) training and progress accuracy of proposed model without dropout regularization.

**Figure 14 bioengineering-09-00152-f014:**
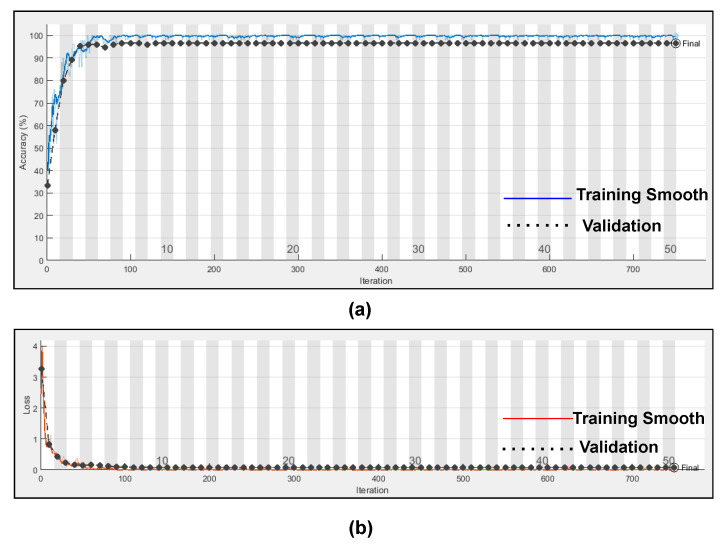
(**a**) Training progress and validation accuracy of proposed model using dropout regularization (**b**) Training progress and validation loss of proposed model using dropout regularization.

**Figure 15 bioengineering-09-00152-f015:**
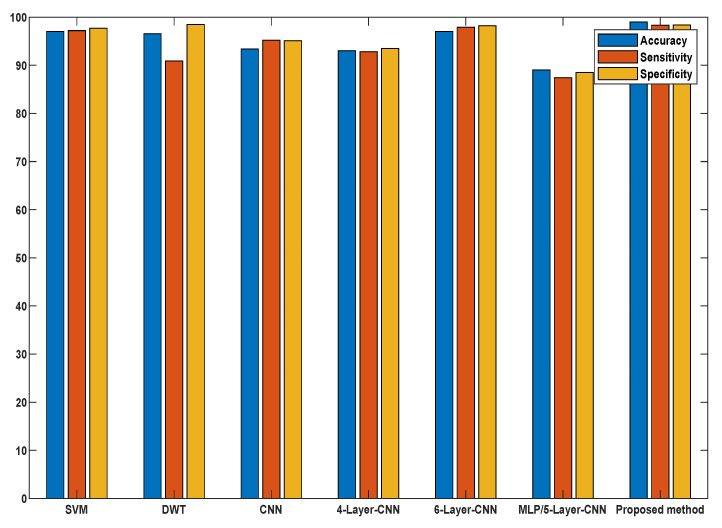
Comparative analysis of the proposed model with the existing sate-of-the-art methods.

**Table 1 bioengineering-09-00152-t001:** Description of all arrhythmia databases from Physionet.

Database Type	No of Recording	No of Samples	Sampling Rate
MIT-BIH Cardiac Arrhythmias (ARR)	96	96 × 65,536	128
BIDMC congestive heart failure (CHF)	30	30 × 65,536	250
MIT-BIH Normal Sinus Rhythm (NSR)	36	36 × 65,536	128

**Table 2 bioengineering-09-00152-t002:** Representation of Complete Layer Architecture of Proposed Model [59].

No. of Layers	Name	Kernel-Size	Filter-Stride	Kernel-Type	No of Neurons
1	Input image	-	-	-	227×227×3
2	Sequence folding layer	-	-	-	227×227×1
3	Conv2D+ReLU+Normalization	[3 3]	1	64	227×227×1
4	Pooling	[2 2]	2	-	227×227×64
5	Conv2D+ReLU+Normalization	[1 1]	1	128	64×64×64
6	Pooling+padding	[2 2], [1 1 1 1]	2	128	64×64×128
7	Conv2D’	[3 3]	1	256	32×32×128
8	Pooling	[2 2]	2	-	32×32×256
9	Sequence unfolding layer	-	-	-	32×32×256
10	flattening	-	-	-	192×192
11	LSTM	-	-	-	4096
12	Fully Connected	-	-	-	4096
13	ReLU	-	-	-	-
14	dropout	-	-	-	-
15	Fully Connected	-	-	-	4096 fully connected layer
16	ReLU	-	-	-	-
17	Dropout	-	-	-	50% dropout
18	Fully Connected	-	-	-	1000 fully connected layer
19	Softmax	-	-	-	-
20	Classification Output	-	-	-	-

**Table 3 bioengineering-09-00152-t003:** Representation of optimised hyperparameters with different test-scores using Grid Search Optimization.

Learn-Rate	Batch-Size	Hidden-Units	Epochs	Mean-Test-Score
0.01	10	32	20	92.35
0.04	64	64	50	83.63
0.05	32	32	30	78.5

**Table 4 bioengineering-09-00152-t004:** Accuracy Measures.

Measures	Formula
Accuracy (*A*)	$A=\frac{\left(TP+TN\right)}{\mathrm{Total}\;\mathrm{no}\;\mathrm{of}\;\mathrm{samples}}$
Precision (*P*)	P=TP/(TP+FP)
Recall (*R*)	$R=(\frac{TP}{TP+FN})$
F-Measure	$F=2*\frac{\left(P*R\right)}{\left(P+R\right)}$
Sensitivity	$\frac{TP}{\left(TP+FN\right)}$
Specificity	$\frac{TN}{\left(TN+FP\right)}$

**Table 5 bioengineering-09-00152-t005:** The table represents overall accuracy, *TP*, *TN*, *FP*, *FN*, F1, Sensitivity (Se), and specificity (Sp) scores for the 2DCNN-LSTM model without k-fold validation.

S. No	Class	Accu.	*TP*	*TN*	*FP*	*FN*	Prec.	Recall	F1 Score	Se	Sp
1	ARR	98%	62	100	0	2	0.97	0.966	0.98	0.96	1
2	CHF	77%	50	113	50	0	0.5	1	0.66	1	0.69
3	NSR	99%	50	113	1	0	0.98	1	0.66	1	0.98

**Table 6 bioengineering-09-00152-t006:** The table represents overall accuracy, *TP*, *TN*, *FP*, *FN*, F1, Sensitivity (Se), and specificity (Sp) score for 2DCNN-LSTM model with k-fold validation.

S. No	Class	Accu.	*TP*	*TN*	*FP*	*FN*	Prec.	Recall	F1 Score	Se	Sp
1	ARR	98.7%	65	98	2	1	1	0.98	0.98	0.98	0.98
2	CHF	99%	48	115	1	2	0.97	0.96	0.96	0.96	0.99
3	NSR	99%	48	115	1	1	0.97	1	0.98	0.97	0.99

**Table 7 bioengineering-09-00152-t007:** Average performance analysis of two experiments for the classification of Arrhythmias.

Experiments	Schemes	Avg-Accuracy	Avg-Specificity	Avg-Sensitivity
A	without dropout regularization	99.8%	99.78%	99.77%
B	with dropout regularization	99%	98.35%	98.33%

**Table 8 bioengineering-09-00152-t008:** Comparison table of the proposed architecture with some other existing architectures in terms of feature extraction (FE) methods, methodology, accuracy, and other statistical classification.

Author	FE	Database	Model	ACC	TPR	TNR	FPR	FNR	Sensitivity	Specificity
Acharya et al. [26]	RP	MIT-BIH arrhythmia database	11-layer CNN	96%	96	95	3	1	95.49%	94.19%
Zengh et al. [64]	CNN	MIT-BIH arrhythmia database	CNN-LSTM	98%	10	11	1	2	97.87%	98.57%
Dinesh et al. [65]	Wavelet-intervals	MIT-BIH Arrhythmia Database	optimized CNN	93.19%	10	11	1	2	93.98%	95%
Oliver et al. [66]	RR-intervals	ECG signal Database (Shaoxing Hospital Zhejiang University School of Medicine)	CNN	93.19%	12	9	12	1	95%	94.30%
Lu et al. [67]	CNN	MIT-BIH Arrhythmia Database	Depth wise seprable CNN with focal loss	98.55%	10	12	1	2	82%	79%
Wang et al. [68]	CNN	MIT-BIH, INCARTDB and SVDB	Depth wise seprable CNN with focal loss	97.40%	11	12	2	1	96.7%	97.8%
Bhekumuzi et al. [69]	2d images using 2D recurrent plots	MIT-BIH Atrial Fibrillation Database, MIT-BIH Malignant Ventricular Ectopy Databas	CNN	95.30%	10	11	2	1	94.2%	95%
Yang et al. [70]	CNN	first China ECG Intelligence Challenge	cascaded CNN	86.50%	13	9	2	1	85.3%	82%
Rahul et al. [71]	RR-intervals	IT-BIH Arrhythmia Database	SVM	99.51%	10	11	1	2	99.28%	99.63%
Chouhan et al. [72]	CNN	IT-BIH Arrhythmia Database	CNN	97.16%	50	100	1	0	99.28%	99.63%
Ullah et al. [38]	CNN	IT-BIH Arrhythmia Database	CNN	97.38%	63	89	1	0	2%	95.63%

## Data Availability

Not applicable.

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
