# Peer review of "A Hybrid Deep Learning Approach for ECG-Based Arrhythmia Classification"

_bioengineering, 2022, doi:10.3390/bioengineering9040152_

Round 1

Reviewer 1 Report

This paper proposed a hybrid deep learning model for ECG-based arrhythmia classification problem. The main contribution is converting 1d ECG signals to 2D scalogram images and applying 2D convolutional neural networks and LSTM network on these images. 

Overall paper structure and writing are great. The problem, data and methods are clearly explained. 

Reviewer 2 Report

The paper presented a system for the classification of arrhythmias by using a CNN-LSTM 
model. A robust approach has been introduced where 2D scalogram images of ECG signals 
are trained over CNN-LSTM model. It is an interesting paper, however, the paper can be 
improved by considering these points….
1. Discussion Section: The authors should include a discussion section mentioning the 
critical findings, strengths, weaknesses, and extensions. A discussion is needed around 
CWT.
2. Recent Publications doing the same strategy of “CNN+LSTM”. It would be good to 
include them and discuss these as well. How Scalogram as intermediate stage helps in 
classification.
Classification of Mental Stress Using CNN-LSTM Algorithms with
Electrocardiogram Signals
Mingu Kang, Siho Shin, Jaehyo Jung , and Youn Tae Kim
Automatic Detection of Atrial Fibrillation Based on
CNN-LSTM and Shortcut Connection
Yongjie Ping 1, Chao Chen 2 , Lu Wu 1,2, Yinglong Wang 2 and Minglei Shu 2,*
Classification of Arrhythmia in Heartbeat Detection Using
Deep Learning
Wusat Ullah,1 Imran Siddique ,2 Rana Muhammad Zulqarnain ,3
Mohammad Mahtab Alam ,4 Irfan Ahmad,5 and Usman Ahmad Raza6
3. Speed of the system: There is no mention of the training time.
4. Memorization vs. Generalization: It is hard to tell if the system is generalizing or 
memorizing? There is no power analysis conducted to tell the # of samples is under or 
over. Some discussion has to be added regarding the sample size discussions.
5. Scientific Validation: There is no scientific validation of this strategy.

Reviewer 3 Report

I have read the manuscript. It is an interesting work, but it presents some issues that have to be solved. Some comments:

  • Introduction should clearly state the aim of the paper.
  • I suggest reporting the statistical matrices on the method section.
  • I suggest reporting the confusion matrix only one time, or as figure or as table.
  • It is not clear how authors tested their method. Did they apply a cross validation method or they evaluated the method on a testing dataset?

Reviewer 4 Report

The manuscript deals with the classification of ECG signals in 3 classes – cardiac arrhythmia (ARR), congestive heart failure (CHF) and normal sinus rhythm (NSR). The ECG signals are initially converted to 2D scalogram via continuous wavelet transformation (CWT) and are supplied to a deep neural network model including CNN and LSTM layers.  

General comments:

The manuscript is not well written and the information in the sections is not well structured. It presents a lot of well-known information but does not present crucial details of the presented study. Another general requirement is the need for adequate language editing.

Specific remarks, questions and recommendations:

  • Sections ‘Introduction’ and ‘Related work’
  • Sections ‘Introduction’ and ‘Related work’ could be joined under the common name ‘Introduction’ and this new section should finish with the aim of the study. The joining should not be mechanically (i.e. just sticking them together), since currently these 2 sections are not well structured and are written chaotically. The reader could not follow the idea in its progress, but is covered with some data for some studies that are not grouped neither by the classification task, nor by applied methodology, nor in some other way. The future section ‘Introduction’ should be fully rewritten.   
  • Subsection ‘1.1. ECG signals’ provides basic information for the ECG that could be found in any ECG related textbook. It is redundant for a scientific paper.
  • If used somewhere in the new vestion of the Introduction, the authors should correct “precocious leads (I, II, III, aVL, aVR, aVF)” to “precordial leads (V1, V2, V3, V4, V5, V6)”.
  • Correct “fetal problems” to “fatal problems”.
  • Current section ‘Related work’ starts very similar to the current ‘Introduction’. This should be avoided and the information should be merged.
  • Recently (2019-2021), the application of CNNs for detection of life-threatening cardiac arrhythmias has been announced and this is worth to be mentioned (Jekova et al, 2021, “Optimization of End-to-End Convolutional Neural Networks for Analysis of Out-of-Hospital Cardiac Arrest Rhythms during Cardiopulmonary Resuscitation”; Krasteva et al, 2020, “Fully Convolutional Deep Neural Networks with Optimized Hyperparameters for Detection of Shockable and Non-shockable rhythms”; Elola et al, 2019, “Deep Neural Networks for ECG-Based Pulse Detection during Out-of-Hospital Cardiac Arrest”).
  • At the end of section ‘Related work’ the authors have stated: “we summarize that our model is more convenient because instead of using 1-D ECG signals as input, we also use 2-D colored Scalogram images as input which are the size of 227 × 227 × 3”. How did the authors reached to this conclusion? Moreover, this is the place to set the aims of the study, not to summarize the conclusions.
  • Section ‘Methodology’
  • The description of the dataset should be extended to present duration of the ECG recordings, sampling rate, amplitude resolution, filtration, etc. This data should be revealed for all subsets of data such as it is done now for BIDMC dataset. The presentation “1:96”, “97:126” and “127”162” is not clear and should be rewritten, e.g. ARR are from the 1st to the 96th row of the array …, etc.
  • Figure 2 is not very informative in its present form. Instead of showing simple examples for the difference between the 3 classes it is recommendable to show also the CWT for these 3 ECG signals in order to present their difference after the applied transformation.
  • In the description of the data it should be written if the data is separated for training, validation and testing and how.
  • The data segmentation is not clearly described. In fact, is it true that the authors have used 90 out of 162 ECG recordings and have extracted 10 segments per ECG recording (these segments are again referred as “recordings” in the text, which is not correct)? How did the authors select 30 out of 36 NSR and 30 out of 96 ARR? What selection criteria did they apply? How are the training and test ECG segments divided? What stands behind [65536/500]=131 and why is it applied? Why 10 out of 131 are selected?
  • “As seen in Fig. 3, where 900 Scalogram images are used as input parameters, with 750 images used for training and for testing 250 images were used after data acquisition and transition, these input parameters are passed through CNN-LSTM architecture for classification of arrhythmias” This sentence is not clear. What data acquisition and transition? How is the image conversion related to Fig. 3? Again the information is not presented in an ordered way.
  • In subsection 3.2.3 there is a lot of redundant information describing basic definitions and at the same time there is lack of information what exactly is done in the study. Researchers who would read the paper know what is CWT and therefore the authors should focus on its particular application in their study.
  • “…this study considers the CWT as a potential candidate solution” – are there other solutions considered in this study?
  • Figure 3 – “Training and validation set” are not defined. If separate validation set is available (as stated on the left of the check-box ‘CNN-LSTM’, why 10-fold cross validation is applied (on the right). Moreover, why ‘CNN-LSTM’ is placed in a check-box but not in an ordinary one?
  • “Twelve bandpass filters per octave are used for CWT, as shown in Fig. 6.” – This is not shown in the figure.
  • The architecture in fig. 7 is not clear. It is hard to follow its correspondence to Table 2. The presentation needs to be improved in this sense. The same is valid for Table 2. Just an example, but not the only unclear point – isn’t the classification output with 3 neurons instead of 4096 as stated in Table 2? In figure 7 – what does it mean 2 groups of 192 or 2 groups of 128?
  • In 3.3.1 and 3.3.2 the authors again have described well-known principles of the used methodology instead of presenting what exactly they have done and why.
  • “The shortcomings of traditional artificial neural network whose output does not depend on its input and just used either for classification and regression purpose [58,59].” It is not clear what exactly are the shortcomings. Should be rewritten.
  • Figures 8 and 9 do not present information that is specific for the methodology elaborated by the authors and presented in this study.
  • How are the hyperparameters of the presented network selected (optimized)?
  • Section ‘Experimental Results and Performance Evaluation’
  • “In this experiment, the dataset is divided into 70% for training and 30% for testing.” – Do the authors mean real independent testing or validation? The last (validation) is done during the training process and is not an independent assessment of the accuracy.
  • The accuracy metrics should be defined (Table 3) and calculated (Table 4) per each of the 3 analyzed classes, i.e. Se(ARR)=TP(ARR)/(TP(ARR)+FN(ARR)), etc.
  • “True Positive (TP) represents the number of positive patients who have been assessed as positive. The True Negative (TN) represents the number of patients who are anticipated to be negative.” Positive or negative when referenced to what? A patient could be positive for ARR and negative for CHF. The description needs to be refined.
  • Results on an independent test dataset must be reported separate for each disease and not as average accuracy, sensitivity and specificity.
  • What are TPRate, TNRate, FPRate, FNRate in Table 5? Again in Table 1 – the F1-score is always in the range [0,1]. How are the values in the table calculated?
  • The confusion matrix in Fig. 12 is built for 148 cases. The used data is for 162 cases. It is not clear what do these 148 cases represent – it is neither the validation set, nor the training set. In fact, the confusion matrix should be presented for a test set, which is not available.
  • 12 and Table 6 represent one and the same information.
  • Table 7 and Fig. 13 – for fair comparison the cited studies should treat the same diseases. Also, it should be mentioned on what dataset are done the tests.

Round 2

Reviewer 4 Report

The authors have considered the remarks and recommendations in my first report. The manuscript is significantly improved. Before publication the authors should take care of the following points:

  • Line 130: Correct “krasteva et al” to “Krasteva et al”
  • The title of the subsection “1.1. motivation and contribution” at the end of section Introduction is not necessary. Moreover, there aren’t other subsections in this section.
  • Correct the title of Table 2 on page 15.
  • Figures 6 and 8 have one and the same caption, which is not a good practice (at least they do not show one and the same thing). I suggest adequate modification of the caption of Figure 6 – e.g. Block-diagram of the proposed method.
  • I recommend section ‘Discussion’ to be separate section but not subsection (3.1) of the results.
  • All figures and tables should be placed in the section where they are referred for the first time. The place of figures 12, 13, 14, 15 and table 8 is definitely not in section ‘Conclusions’.
  • The authors should refer to the tables and figures in the text in one and the same way – i.e. use either Fig. or Figure (table or Table) but not both.
